# Maggot extract accelerates skin wound healing of diabetic rats via enhancing STAT3 signaling

Mo-Li Wu[1]☯, Zhe-Ming Yang[1]☯, Hai-Chao Dong[2], Hong Zhang[2], Xu Zheng[1], Bo Yuan[1,2], Yang Yang[1], Jia Liu◉[1]*, Pei-Nan Li[2]*

1 Department of Cell Biology, College of Basic Medical Sciences, Dalian Medical University, Dalian, China,
2 Department of Orthopedic Surgery, Second Affiliated Hospital, Dalian Medical University, Dalian, China

☯ These authors contributed equally to this work.
* jialiudl2022@163.com (JL); lipndl@163.com (PNL)

**Data Availability Statement:** All relevant data are within the manuscript and its Supporting Information files.

## Abstract

### Background

Diabetic skin wound is a complex problem due to the disruption of normal repairing program and lack of effective remedy. *Lucilia sericata* larvae (maggot) is a folk method to treat chronic skin wound, while its therapeutic effects on that caused by diabetic remains unknown.

### Objective

This study aims to investigate the therapeutic effects of maggot extract (M.E.) on diabetic skin wound and its molecular mechanism by establishing the skin wound model of diabetic Sprague Dawley (SD) rats.

### Methods

Diabetic model was established by injecting intraperitoneally streptozotocin in SD rats under specific pathogen-free (SPF) conditions. The rat fasting blood glucose values were ≧16.7 mmol/L 72 hours after intraperitoneal streptozotocin (60mg/kg body weight) injection. The rats were divided into five groups (*n* = 10/group): normal group: normal SD rats without any treatment, diabetic blank group: the diabetic rats without any treatment, Vaseline group: the diabetic rats dressed with Vaseline, recombinant human epidermal-growth-factor (rhEGF) group: the diabetic rats dressed with a mixture of Vaseline and 200 μg/g rhEGF, M.E. group: the diabetic rats dressed with a mixture of Vaseline and 150 μg/ml maggot extract. The round open wounds (1.0 cm in diameter) down to the muscle fascia were made on both sides of rat dorsa by removing the skin layer (epidermis and dermis) and were daily photographed for calculating their healing rates. Hematoxylin-eosin (HE) and Masson's trichrome staining were performed on skin wound sections to analyze re-epithelialization and granulation tissue formation. Immunohistochemical (IHC), immunofluorescent (IF) stainings and Western blotting were conducted to analyze the statuses of STAT3 signaling.

**Funding:** This work was supported by the Grants from National Natural Science Foundation of China [Nos. 81450016 and 81272786 to J.L.] and Natural Science Foundation of Liaoning Province, China [No. 2019-ZD-0650 to P.N. L.]. The funders have roles in study design, data collection and analysis, decision to publish, or preparation of the manuscript.

**Competing interests:** The authors have declared that no competing interests exist.

## Results

The average wound healing rates on the 14th day were 91.7% in the normal, 79.6% in M.E., 71% in rhEGF, 55.1% in vaseline and 43.3% in the diabetes blank group. Morphological staining showed more active granulation tissue formation, re-epithelialization and neovascularization in M.E.-group than those in the blank and the vaseline-treated groups. Decreased p-STAT3 nuclear tranlocation and down-regulated Bcl-2, CyclinD1 and vascular endothelial growth factor (VEGF) expression were evidenced in the diabetic rats, which could be improved by rhEGF and especially M.E.

## Conclusion

Maggot extract would be an alternative and/or adjuvant candidate for the better management of diabetic skin wounds because of its activity in enhancing STAT3 activation.

## Introduction

Diabetes is a chronic metabolic disease, which has become a growing international health concern [1,2]. Diabetes (type 1 and 2) involves many complications, and the most common one is the delayed healing and even nonunion of acute and chronic skin wounds [3]. Multiple factors or molecular machinaries responsible for wound healing are altered in diabetic individuals, leading to the reduced regenerative activity of skin wounds. It has been proposed that the local peripheral nerve and vascular alterations are the main reasons for the difficulty of skin wound healing [4]. Current options for wound management of diabetic patients are: 1) strict control of blood glucose; 2) application of vasodilators to improve microvascular circulation; 3) antibacterial and anti-inflammatory treatments; 4) surgical debridement and 5) local nursing and health care [5]. Although the above measures include both the whole and local treatments, the incidence of diabetic wounds and the disability rate caused by them remain almost unchanged, indicating the unsatisfactory efficacy of the above approaches [6]. Therefore, it would be necessary to alleviate the influence of pathogenic factors and meanwhile to accelerate the wound healing. Because of the complexity of the pathogenic factors of diabetic wounds, the new agent (drug) containing beneficial components for neovascularization and cell proliferation would be helpful.

*Lucilia sericata* (Meigen 1826, Diptera: Calliphoridae) is a widespread blow fly species. The medicinal value of *Lucilia sericata* larvae (maggot) has long been recognized [7]. The maggots are traditionally used to remove the necrotic tissue in the deep wound where the conventional surgery fails to reach [8]. In addition, the maggot body has a strong self-repairing ability after injury, because of its richness in regeneration promoting elements [9]. Even though, the traditional dry baking method significantly reduced the bio-activity of the maggot product [10]. For this reason, we had prepared fresh maggot extract and shown its beneficial effects on normal rat skin wound healing [11]. Moreover, the enhanced signal transducer and activator of transcription 3 (STAT3) activation and upregulation of the anti-apoptotic gene B-cell lymphoma-2 (Bcl-2) were found in maggot extract treated skin wounds in normal rats [11]. Because activated STAT3 signaling controls proliferation and differentiation during wound healing and skin remodeling [12], the above results provide scientific evidence for the effectiveness of maggot extract in improving wound healing [13].

It has to be pointed out that the results so far obtained from our previous study are all from the healthy experimental animals and the skin wounds of those animals are repairable as a matter of course. Alternatively, the maggot extract merely shortens the healing time to a certain extent. Therefore, the key point of the medication value of the maggot extract should be its therapeutic efficacy for the refractory wounds such as diabetic skin wounds. It has been evidenced that the remarkable reduction of phospho-STAT3 (p-STAT3) immunopositive keratinocytes and down-regulation of some growth factors such as interleukin-22 (IL-22) and forkhead box M1 (FOXM1) are responsible for the delayed wound healing of diabetic wounds [14,15]. For these reasons, we consider that maggot extract might also exert beneficial effects on diabetic rat skin wound healing via improving the activity of STAT3 signaling. The current study aims to address the therapeutic effects of maggot extract on diabetic skin wound and its molecular mechanism.

## Materials and methods

### Ethics approval

This research project was approved by Animal Care and Use Committee of Dalian Medical University (DMU) in the ethics approval number AEE18055 to guarantee that the number of experimental animals should be restrictively controlled and all works involving experimental animals were performed in full compliance with NIH (National Institutes of Health, USA) Guidelines for the Care and Use of Laboratory Animals.

### Establishment of type 1 diabetic rat model

50 male Sprague Dawley (SD) rats (aged 5–6 weeks, 200–220 g), were provided by SPF (specific pathogen free) experimental animal center of Dalian Medical University. 4 mg/ml streptozotocin (Sigma-Aldrich, St. Louis, MO, USA) in citric acid/sodium citrate buffer solution was prepared for injection by the method described elsewhere [16,17]. All rats were fed with standard diet under 20–26˚C. After two weeks of adaptive feeding, 40 rats were randomly selected for inducing diabetic model. Briefly, the rats were fasted for 12 hours, followed by one-time intraperitoneal injection of 60 mg/kg streptozotocin. Rats were fasted for 12 hours after injection. 72 hours after streptozotocin injection, the fasting tail venous blood of the treated rats was sampled and checked by blood glucose meter (Accu-chek active; Roche). The rats were diagnosed with diabetes when their fasting blood glucose levels were over 16.7 mmol/L [18].

### Maggot extract preparation

*Lucilia sericata* blowflies were obtained from BIOWIM Technology Development Co., Ltd (Dalian, China) and maintained in a closed container with a net at a constant temperature of about 22˚C. Their larvae were collected when they were reared to later-second or early-third stages. The collected maggots were rinsed three times with autoclaved pure water, placed in 3.5% formaldehyde normal saline solution for 5 minutes, in 2% $H_2O_2$ for 3 minutes, in 1% hydrochloride acid for 5 minutes and finally subjected to three washes with autoclaved pure water. The collected maggots were directly snap frozen in liquid nitrogen and stored at −80˚C until extract preparation. The extracts of maggots were prepared by series sectioning the frozen maggot tissues into pieces (5 μm in thickness) and then putting the frozen sections into the test tubes containing pH7.5 phosphate buffered saline (PBS). By this way, the high-quality maggot extracts were prepared in high yields [11]. As a comparison, maggot extract was treated via the traditional method. Briefly, 400 mg of the frozen maggots were weighed. One half (200mg) of them was sectioned into pieces (5 μm in thickness) and then put into ice-cold

lysis buffer (Beyotime P0013B), and the other half was dried at 35˚C for 12 hours by the method described elsewhere [19], smashed them into powder and put the powder into the same volume (200 μl) ice-cold lysis buffer. The same volume of the protein samples exacted by the two methods were separated by 10% sodium dodecylsulfatepolyacrylamide gel electrophoresis (SDS-PAGE).

## Skin wound model and treatments

10 normal and 40 diabetic rats were anesthetized with ketamine (75mg/kg body weight) and xylazine (10mg/kg body weight) intraperitonealy. The hair in the dorsal skin of rats was shaved and the round open wounds (1.0 cm in diameter) down to the muscle fascia were made on both sides of rat dorsa by removing the skin layer (epidermis and dermis). All of the operated rats were separately fed (one rat/cage). 50 SD rats were divided into 5 experimental groups (10 rats/group): Group 1, normal SD rats without any treatment (normal group); Group 2, the diabetic rats without any treatment (diabetic blank group); Group 3, diabetic rats dressed with Vaseline (Rhawn, Shanghai, China, Vaseline group) [11]; Group 4, diabetic rats dressed with a mixture of Vaseline [11] and 200 μg/g recombinant epidermal growth factor (Pavay Gene Pharmaceutical, Guilin, China, rhEGF group); Group 5, the diabetic rats dressed with a mixture of Vaseline [11] and 150 μg/ml maggot extract (M.E. group). The experimental treatments lasted for 14 days by daily dressing the reagents mentioned above. Maggot extract was quantified by Ultraviolet spectrophotometer-based calculation of protein concentration.

## Wound examination

The wound areas were measured and the wound healing rates were calculated to evaluate the wound healing in different experimental groups. The status of wound healing and the contraction of wound regions were observed and photographed. The electronic images were saved to computer and the wound areas recorded in them were measured by Image J software (National Institutes of Health, USA) with the method described elsewhere [20]. The average wound healing rate in each of the experimental groups was calculated with the formula: wound healing rate = (initial wound area—uninformed area) / initial wound area × 100%.

## Wound edge biopsy

The rats were anesthetized and sacrificed through cervical dislocation performed by appropriately trained and competent personnel at the end of the experiment. The skin tissues in the size of $0.3 \times 0.3 \times 0.2$ cm were biopsied from the wound margins of the five experimental groups. Half of the tissues was snap-frozen in liquid nitrogen, and then moved to the refrigerator at—80˚C for later molecular analyses. The remaining part of the tissues was fixed in 10% neutral formalin buffer to prepare paraffin-embedded tissue block for histological, immunohistochemical and immunofluorescent stainings.

## Histological, immunohistochemical and immunofluorescent analyses

Hematoxylin-eosin (HE) and Masson's trichrome staining (Solarbio Science & Technology Co. Ltd, Beijing, China) were performed on 4 mm-thick central wound sections to analyze re-epithelialization and granulation tissue formation, respectively. Immunohistochemical (IHC) and immunofluorescent (IF) staining were conducted on wound tissue sections by the method described elsewhere [21]. The antibodies used were: Bcl-2 (bs-0032R, Bioss Inc., Beijing, China; 1:250); STAT3 (sc-8019, Santa Cruz Inc., USA; 1:200); VEGF (bs-1313R, Bioss Inc., Beijing, China; 1:200); CyclinD1 (bs-20596R, Bioss Inc., Beijing, China; 1:250); p-STAT3 (sc-

8059, Santa Cruz Inc., USA; 1:200). The color reaction was performed by using 3, 3'-diamino-benzidine tetrahydrochloride (DAB) after the binding of the primary antibody (Vector Laboratories, Burlingame, CA, USA). For double IF staining, the tissue sections were washed with pH 7.4 PBS, incubated in 3% $H_2O_2$ for 10 min and then with the first antibody at 4°C overnight in a humid chamber. Finally, the tissues were co-incubated with FITC-conjugated goat anti-mouse IgG and PE-conjugated goat anti-rabbit IgG (both 1:100; Santa Cruz Biotech, Santa Cruz, CA, USA) at 37°C for 60 min in the dark, sealed with fluorescence mounting medium, and observed and imaged under a fluorescence microscope (BX53F, Olympus, Tokyo, Japan).

## Protein extraction and Western blotting

Western blotting was conducted using antibodies against Bcl-2 (bs-0032R, Bioss Inc., Beijing, China; 1:500); STAT3 (sc-8019, Santa Cruz Inc., USA; 1:600); VEGF (bs-1313R, Bioss Inc., Beijing, China; 1:500); CyclinD1 antibody (bs-20596R, Bioss Inc., Beijing, China; 1:500); p-STAT3 (sc-8059, Santa Cruz Inc., USA; 1:500), respectively. The experiment was performed by the method described elsewhere [11]. Briefly, the sample proteins (20μg/well) were separated by 10% SDS-PAGE and transferred to polyvinylidene difluoride membrane (Amersham, Buckin ghamshire, UK). The membrane was blocked in 5% skimmed milk (Sigma-Aldrich) Tris-buffered saline (TBS-T) (10 mmol/L Tris–HCl, pH 8.0, 0.5% Tween 20 and 150 mmol/L NaCl) at 4°C. After three washes with TBS-T, the membrane was incubated for 3h with the primary antibody at room temperature. Appropriate secondary HRP-conjugated antibodies were used for detection with chemiluminescent ECL reagents (Roche GmbH, Mannheim,Germany). In addition, an antibody against β-actin (ProteinTech 66009–1) was used as a loading control. The labeling signal was removed with a stripping buffer (62.5 mmol/L Tris–HCl, pH 6.7, 100 mmol/L β-mercaptoethanol, 2% sodium dodecyl sulfate (SDS), and the membrane was re-incubated with another antibody until all the parameters were examined.

## Statistical analysis

Statistical analysis was performed using SPSS 21.0 (IBM Corp., Armonk, NY, USA). Data are presented as mean ± standard deviation. Data comparisons among experimental groups were performed using Student's $t$ test and analysis of variance (ANOVA), with a value of $P < 0.05$ being considered as statistically significant.

# Results

## Better quality of fresh maggot extract

Ultraviolet spectrophotometer-based calculation of protein concentration revealed that the protein yield per unit weight of fresh maggots (15.695 μg/mg) is 1.4 folds higher than that (11.545 μg/mg) prepared from the dried maggot powder. The SDS-PAGE result further demonstrated that the protein content and composition of fresh maggot extract (M.E.) is better than that of the maggot powder (Fig 1). The extract prepared from fresh maggots (M.E.) was selected for further experiments.

## Establishment of diabetic rat model

The typical symptoms of diabetes include polydipsia, polyphagia and polyuria, which was observed in rats on the third day after streptozotocin injection. As shown in Fig 2, the diabetic rats but not the normal ones showed weight loss, whereas the fasting blood glucose levels of normal rats were less than 7.0 mmol/L, and in streptozotocin-treated rats were distinctly

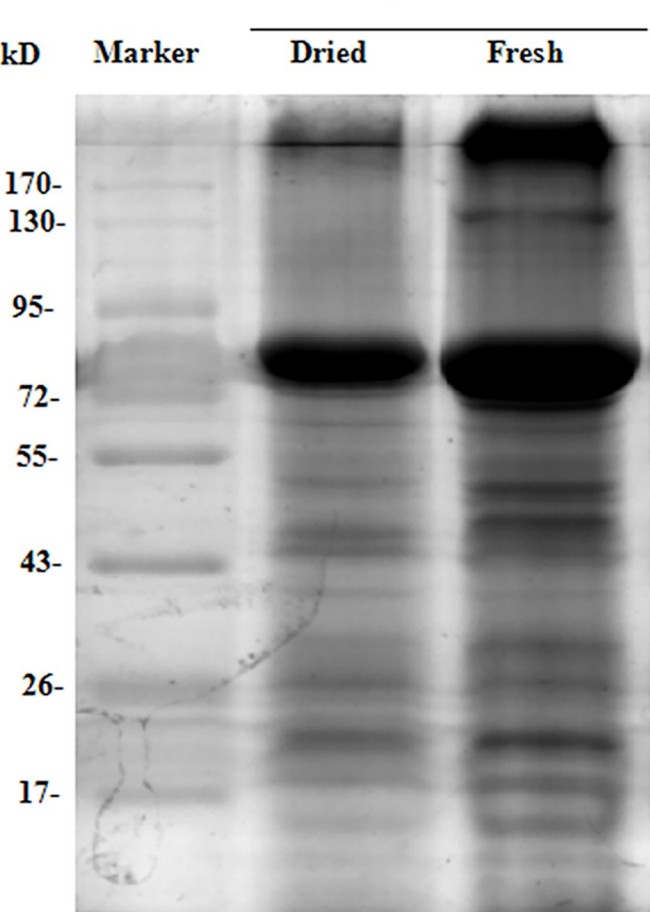

**Fig 1. Comparison of the quality of maggot extracts prepared from the baked maggots and fresh maggots by SDS-PAGE electrophoresis.**

increased (ranging from 25.6 to 27.3 mmol/L). The difference of fasting blood glucose levels between the normal and streptozotocin-treated rats was statistically significant ($P < 0.01$; Table 1).

### Different wound healing rates of experimental groups

Two weeks after the establishment of diabetic model, the skin circular wounds in 1.0 cm diameter were made on both sides of rat dorsa (Fig 3A). The reduction of wound diameter of all groups appeared 3 days after wounding in different speeds. The wound contraction, tissue granulation and epidermal growth of the normal group were faster than those of the diabetic groups. Of the diabetic groups, wound healing efficiency of M.E. group was quicker than rhEGF-treated, vaseline-treated and diabetic blank group (Fig 3B–3D). As shown in Table 2 and Fig 3C, the wound healing rates of normal group (7th day: 47.6% ± 2.8%, 14th day: 91.7% ± 3.7%) were much higher than those of diabetic blank group (7th day: 21.6% ± 2.1%, 14th day: 43.3% ± 2.3%, $P < 0.05$). On the 7th and the 14th day, the wound healing rates of ME group were 38.2% ± 2.5% and 79.6% ± 4.2%, rhEGF group were 33.6% ± 3.1% and 71.0%±4.0%, and vaseline group were 22.8% ± 3.1% and 55.1%±5.0%, respectively. Statistical analyses showed

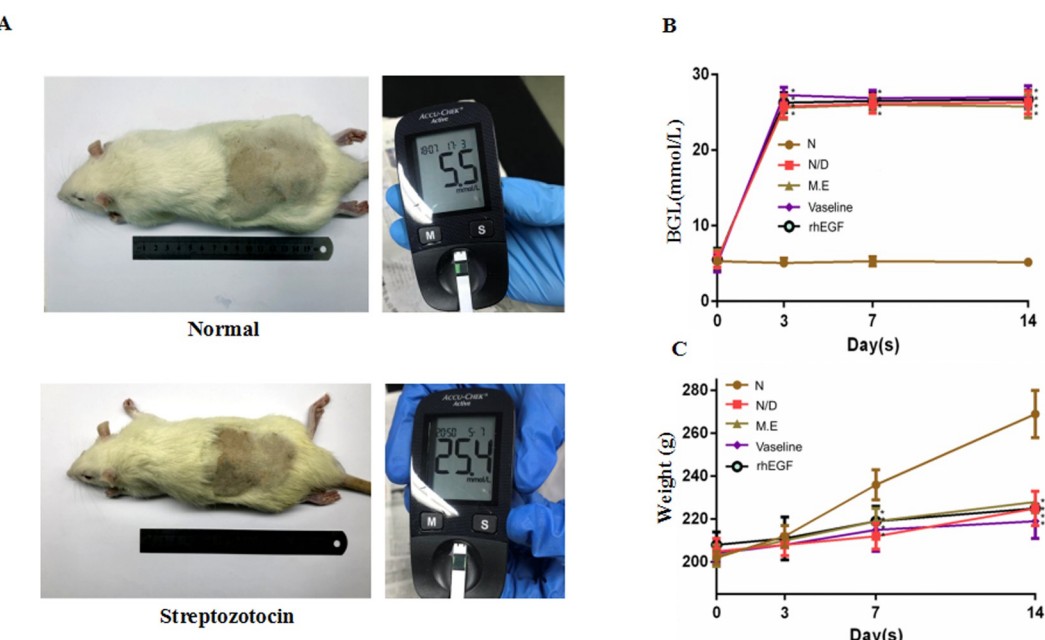

**Fig 2. Establishment of diabetic rat model via streptozotocin injection.** A, The appearance and blood glucose measurement of normal rats and streptozotocin-treated rats. B, The average fasting blood glucose levels (BGL) of SD rats in the 5 groups. C, The average body weights of SD rats in the experimental groups at different time points. The data are presented as mean ± SD. $n$ = 10. *, $P < 0.01$, compared with normal rats.

that wound healing rate of M.E. group and rhEGF group were significantly higher ($P < 0.05$) than that of the vaseline group on the 7th and 14th day after wound modeling.

## Active regeneration of maggot extract- and rhEGF-treated wounds

To investigate wound regeneration, re-epithelialization and granulation tissue formation were checked by HE and Masson's trichrome staining performed on the whole layer skin wound tissues biopsied from each experimental group on the 14th day. As shown in Fig 4, a large number of infiltrating inflammatory cells, disordered tissue architecture, lack of granulation tissue

**Table 1. Average fasting blood glucose levels and body weights of the rats in different groups during the treatments.**

| Model group | Average fasting blood glucose levels (mmol/L) /Average body weight (g) | | | |
|---|---|---|---|---|
| | Before streptozotocin injection | After streptozotocin injection | | |
| | | Day 3 | Day 7 | Day 14 |
| Untreated diabetic rats | 5.5±0.4/205±6 | 25.8±1.7/208±5 | 26.1±1.3/212±6 | 26.3±1.5/225±8 |
| Vaseline-treated | 5.1±0.8/204±4 | 27.3±1.3/208±5 | 26.9±1.8/215±8 | 27.0±1.5/219±8 |
| Maggot extract-treated | 5.6±0.3/203±5 | 25.6±1.7/210±7 | 26.0±1.8/219±6 | 25.8±1.5/228±5 |
| rhEGF-treated | 5.5±0.4/208±6 | 26.3±1.4/211±9 | 26.5±1.3/219±7 | 26.7±1.3/225±8 |
| **Normal group** | **Average fasting blood glucose levels (mmol/L) /Average body weight (g) measured at the same time as the model groups** | | | |
| Normal rats | 5.3±0.5/202±3 | 5.1±0.6*/212±5 | 5.3±0.6*/236±9# | 5.2±0.5*/269±12# |

*, Significant difference ($P < 0.01$) between normal rats and diabetic rats without and with the treatments

#, Significant difference ($P < 0.01$) between normal rats and diabetic rats without and with the treatments; Model group, rats with streptozotocin injection; Normal group, rats without streptozotocin injection.

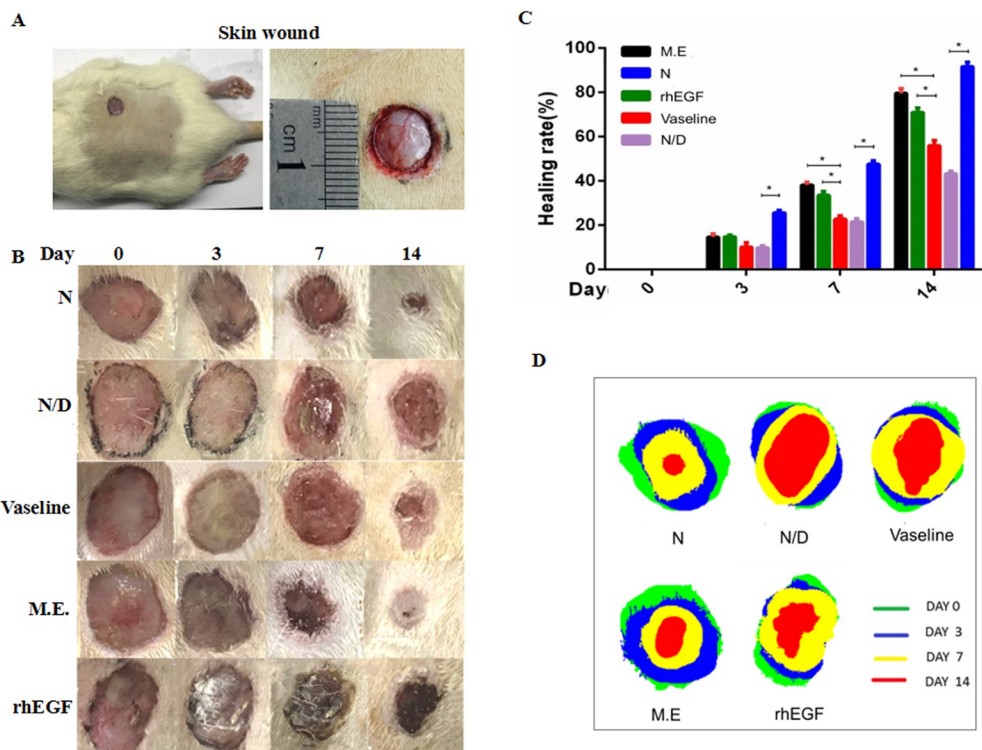

**Fig 3. The effects of different treatments on wound healing at different time points.** A, A round skin wound was made on rat dorsa. B, The wound model rats were divided into 5 experimental groups and their wounds were locally treated by Vaseline, rhEGF gel (rhEGF) and maggot extract (M.E.), respectively. The wounds of normal rats (N) and the diabetic rats (N/D) without treatment were cited as blank controls. C, The wound healing rates of five experimental groups. D, Analogic diagram reflecting the progress of wound healing. The data are shown as mean ± SD, $n = 10$. *, $P < 0.05$.

structure, less neovascularization and incomplete re-epithelialization were found in the diabetic blank group and vaseline-dressed group. In contrast with the control groups, lesser inflammatory cell infiltration, dense granulation tissue structure, abundant collagen fibers as well as capillary formation were observed in the wound tissues of maggot extract-treated and rhEGF-treated groups.

**Table 2. Average wound healing rates of five experimental groups at three time points.**

| Group | Wound healing rates (%, $n = 10$) | | |
|---|---|---|---|
| | Day 3 | Day 7 | Day 14 |
| Normal rats | 25.7±1.9 * | 47.6±2.8 * | 91.7±3.7 * |
| Untreated diabetic rats | 9.9±2.4 | 21.6±2.1 | 43.3±2.3 |
| Vaseline-treated | 10.2±3.6 | 22.8±3.1 | 55.1±5.0 |
| Maggot extract-treated | 14.7±2.1[#&] | 38.2±2.5[#&] | 79.6±4.2[#&] |
| rhEGF-treated | 14.8±2.5[#] | 33.6±3.1[#] | 71.0±4.0[#] |

*, $P < 0.05$ in comparison with untreated diabetic group at the same time

#, $P < 0.05$ in comparison with Vaseline-treated group at the same time

&, $P < 0.05$ in comparison with normal group at the same time.

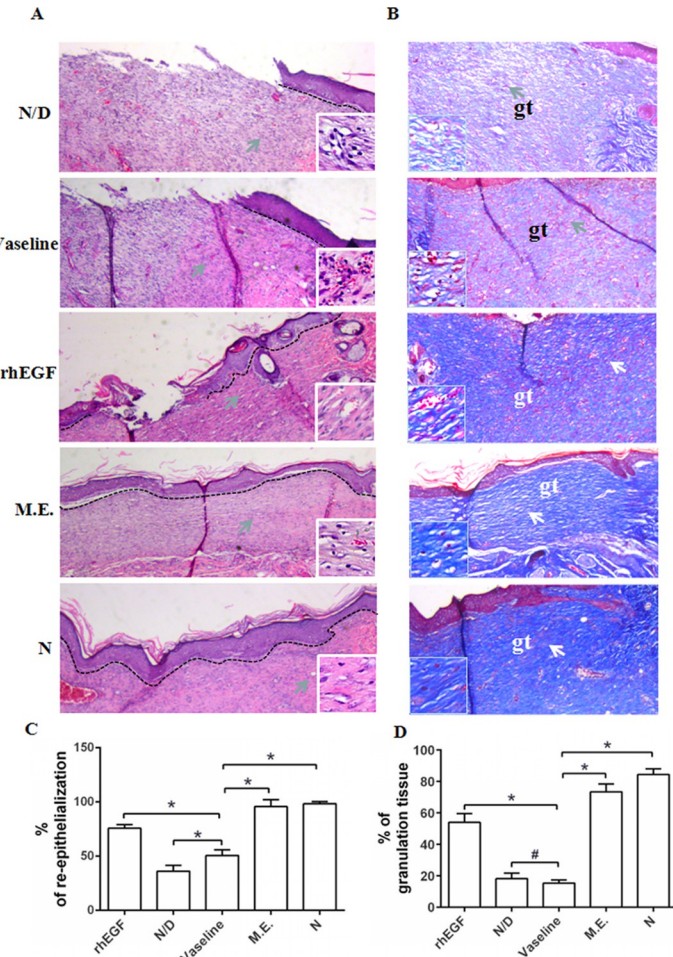

**Fig 4. The morphology analysis of wound healing in different experimental groups on the 14th day of the experiment.** A, Composite images of hematoxilyn/eosin-stained central wound sections of five experimental groups to show re-epithelialization (×50). Epithelial tongues are highlighted by dash line. Insets: the regions in higher magnification (×200). B, Masson's trichrome staining for evaluating granulation tissue (gt) composition (×50). Red color: cytoplasm, muscle fibers; blue color: collagen fibers. Insets: the regions in higher magnification (×200). C and D, Quantitative analysis of re-epithelialization (percentage of distance covered by epidermis) and granulation tissue (percentage of collagen fibers in granulation tissue). *, $P < 0.01$; #, $P > 0.05$.

## Differential STAT3 and its downstream gene expression in normal and diabetic rats

Immunohistochemical (Fig 5A) and immunofluorescent (Fig 5B) staining revealed that STAT3 protein as well as Bcl-2, Cyclin D1 and VEGF were widely distributed in regenerative wound tissues of normal rats, accompanied with frequent p-STAT3 nuclear translocation. In contrast, the expression intensities of the above parameters in untreated or vaseline-dressed wounds of diabetic rats were distinctly weak. The results of Western blotting were in accordance with that of IHC in terms of increased p-STAT3 level and Bcl-2, Cyclin D1 and VEGF expression in the wound tissues of normal rather than the diabetic rats (Fig 5C; $P < 0.05$).

## Enhanced STAT3 activation in maggot extract- and rhEGF-treated tissues

Because recombinant epidermal growth factor (rhEGF) was commonly used in the clinical treatment of skin wound healing, the statuses of STAT3 signaling and Bcl-2, Cyclin D1 and

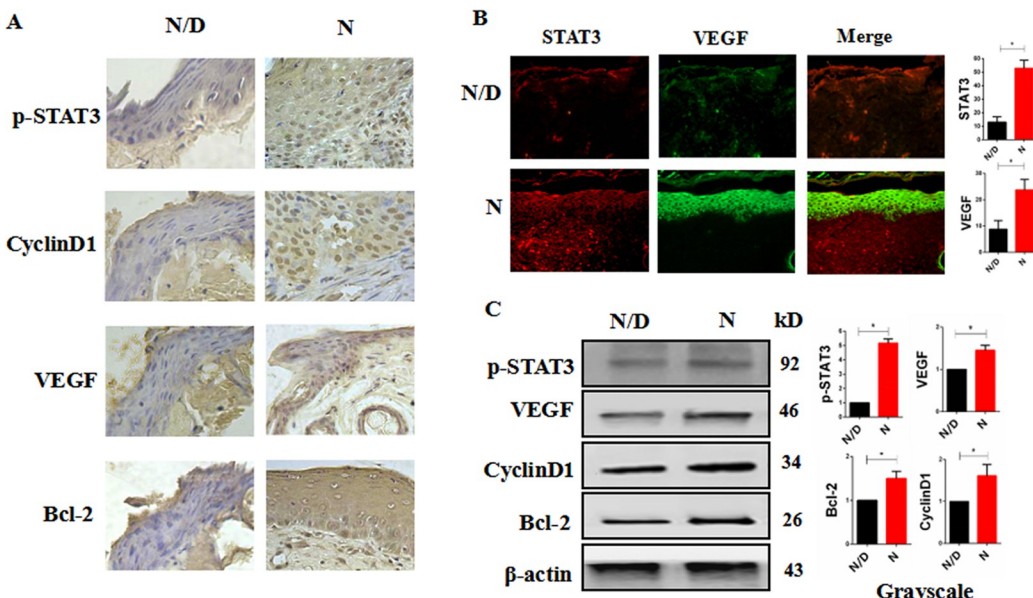

**Fig 5. Distinct STAT3 activation and its downstream gene expression patterns between normal and diabetic rat wound tissues.** Immunohistochemical staining (A) and immunofluorescent labeling (B) were performed on the wound tissues. Western blotting (C) was conducted on the sample proteins prepared from the wound tissues of normal (N) and diabetic rats without any treatments (N/D). The data are shown as mean ± SD. $n = 10$. *, $P < 0.05$.

VEGF expressions in the wounds treated with rhEGF were checked and compared with the results from maggot extract-treated wound tissues. It was found that the immunolabelings of p-STAT3, Bcl-2, Cyclin D1 and VEGF were strongly positive in the wound tissues of diabetic rats treated with maggot extract and rhEGF (Fig 6). Western blotting further demonstrated that Bcl-2, Cyclin D1 and VEGF levels in maggot extract-treated group were higher than those in the vaseline-treated and untreated group while in coherence with those of rhEGF-treated and the normal control group (Fig 6B; $P < 0.05$).

## Discussion

Diabetes is a metabolic disease characterized by elevated blood glucose and its incidence keeps increasing [22]. Instances of chronic, hard-healing, or non-healing diabetic wounds and ulcers are predicted to increase, which seriously affects patients' quality of life. Several standard treatments of diabetic wounds have been proposed, while the effects are limited in terms of accelerating the course of wound healing. For instance, becaplermin (platelet derived factor, PDGF) is the only drug approved by Food and Drug Administration (FDA), USA for the treatment of diabetic foot ulcer, but its efficacy is poor and usually increases the risk of death [23]. This therapeutic dilemma is largely due to the complexity of pathogenesis of diabetic wounds [24]. It would be of clinical significance to explore multipotent novel agent(s) that promotes the diabetic skin wound healing without causing side effects. To fulfill this purpose, we established rat diabetic model by streptozocin injection and made skin wounds on rat dorsa. The results of wound area calculation showed that the skin wound contraction of the untreated diabetic rats was significantly delayed compared with that of the untreated normal ones. The wounds of normal rats were almost healed (healing rates: 91.7% ± 3.7%), while the wounds of the corresponding diabetic group healed partly (healing rates: 43.3% ± 2.3%) on the 14th day. The significant delay of wound healing makes the diabetic rats a suitable model for experimental therapy and investigation of potential agent for better treatment of diabetic skin wounds.

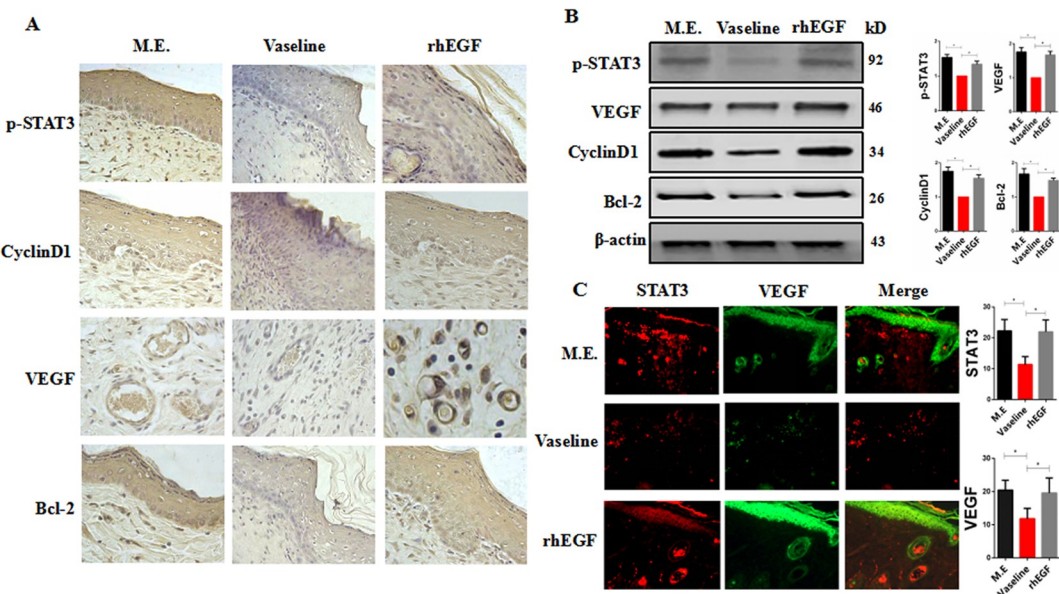

**Fig 6. Enhanced STAT3 activation and upregulated VEGF, Cyclin D1 and Bcl-2 in maggot extract and rhEGF treated wound tissues.** Immunohistochemical stainings (A), Western blottings (B) and immunofluorescent labelings (C) were performed on the samples of the vaseline, maggot extract (M.E.) and rhEGF treated wounds of the diabetic rats. The data are shown as mean ± SD. $n = 10$. *, $P < 0.05$.

Normally, wound healing involves hemostasis, proliferation and remodeling, while this stepwise process becomes disordered due to the lack of wound repair promoters in the diabetic skin [25]. Therefore, the biological effects of new agent should be multifaceted and maggot extract, as a mixture of natural products, is a potential candidate because of its richness in bioactive substances [26]. The roles of maggots in promoting wound healing have been recognized [27,28] and multiple bioactive substances were identified in its extract [9]. However, the traditional dry baking method reduces the components and especially the biological activity of maggot extract. We had therefore developed an unique method by which the maggot extract with high quality and high yield was prepared and its beneficial effects on the skin wound healing of normal adult rats were evidenced [11]. Even though, we proposed that the real clinical medicinal value of the extract is its effectiveness on refractory wounds such as skin injuries in diabetic patients. To address this issue, the influence of maggot extract in diabetic skin wound healing was elucidated by citing Vaseline as a conventional dressing control, rhEGF as the clinical treatment control and the untreated wounds of the normal and the diabetic rats as blank controls. The results showed that on the 14th day of treatments, the wound healing rates of maggot extract-treated (79.6%±4.2%) and rhEGF-treated group (71.0%±4.0%) were similar and faster than those of other groups. Because rhEGF has been commonly used to treat skin wounds, our results indicated that maggot extract may equivalently promote diabetic wound healing. This notion is supported by reduced local inflammation, active neovascularization and accelerated re-epithelization observed in maggot extract-treated wound regions. Further investigation of the underlying mechanism of maggot extract promoted wound healing can provide scientific basis for the practical use of this natural product to treat diabetic wounds.

It is known that trauma as a stimulator can trigger chain reactions of tissue remodeling [29]. In the skin, the re-epithelialization process includes the migration, proliferation and differentiation of keratinocytes, elimination of damaged tissues, and production of extracellular matrix. These events are mainly mediated by extracellular signal transduction pathways in

which STAT3 signaling plays active roles [30] because some of its downstream target genes such as VEGF, CyclinD1 and Bcl-2 are highly expressed in the wound healing area and act as key factors to accelerate wound healing [13]. Our previous works demonstrated that the main mechanism of maggot extract-promoted wound healing of normal rat skin is to increase STAT3 signaling activity and then to upregulate the expression of the above-mentioned damage repair promoting genes [11]. Given this evidence, we speculate that maggot extract may also promote the healing of diabetic skin injury through similar cellular and molecular mechanisms. To address this issue, immunohistochemical staining and Western blot analyses were conducted to check the statuses of STAT3, p-STAT3, VEGF, CyclinD1 and Bcl-2 in the wound tissues of the experimental groups. The results showed that the nuclear translocation of p-STAT3 and VEGF, CyclinD1 and Bcl-2 production in skin wounds of normal rats were significantly higher than that of the diabetic blank and vaseline-treated tissues and this situation can be reversed, at least in large, by rhEGF as well as maggot extract. These results thus confirm the lack of wound repair promoting factors in diabetic skin tissue and suggest that both maggot extract and rhEGF promote diabetic wound healing via similar molecular route, i.e. the enhancement of STAT3 signaling transduction, because the interplay of EGF and STAT3 signaling has been well known [31].

The application of maggot and its extract in wound care keeps increasing and has been proved effective [32]. To the best of our knowledge, the comparison of experimental therapeutic efficacy of maggot extract with that of conventional wound care agent(s) is still lesser reported. Because EGF-containing ointment has been commonly used in clinical management of skin wounds and the data concerning its molecular effects on the wound tissues have been well documented, it was selected as the clinically relevant control. The results show that the promoting effects of maggot extract on wound healing of diabetic rats (79.6% ± 4.2%) is relatively better than that of rhEGF-treated group (71.0%±4.0%), indicating the presence of additional biological effects of maggot extract on the skin wounds beyond stimulating STAT3 activation. Elucidation of the influence(s) of pathway inhibitors such as STAT selective inhibitor AG490 in maggot extract-promoted wound healing of diabetic rat skin would further strengthen this notion.

## Conclusions

The results of current study demonstrate the effectiveness of maggot extract in improving skin wound healing of diabetic rats via promoting the activity of STAT3 signaling and up-regulating the expression of STAT3 downstream target genes Bcl-2, Cyclin D1 and VEGF. In this context, maggot extract would be an ideal agent to resolve the complexity of skin wound healing of diabetic patients.

## Supporting information

**S1 Raw images.**
(PDF)

**S1 Raw data.**
(PDF)

**S2 Raw data.**
(PDF)

**S3 Raw data.**
(PDF)

## Author Contributions

**Conceptualization:** Mo-Li Wu, Jia Liu, Pei-Nan Li.

**Data curation:** Mo-Li Wu, Zhe-Ming Yang, Hong Zhang, Pei-Nan Li.

**Formal analysis:** Mo-Li Wu, Zhe-Ming Yang, Hai-Chao Dong.

**Funding acquisition:** Jia Liu, Pei-Nan Li.

**Investigation:** Mo-Li Wu, Zhe-Ming Yang, Xu Zheng, Bo Yuan, Yang Yang.

**Methodology:** Mo-Li Wu, Zhe-Ming Yang.

**Supervision:** Jia Liu, Pei-Nan Li.

**Writing – original draft:** Mo-Li Wu, Zhe-Ming Yang.

**Writing – review & editing:** Mo-Li Wu, Zhe-Ming Yang, Jia Liu.

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
