## [Decision Letter · Decision Letter 0]

21 May 2024

PONE-D-24-13961Maggot extract accelerates skin wound healing of diabetic rats via enhancing STAT3 signalingPLOS ONE

Dear Dr. Liu,

Thank you for submitting your manuscript to PLOS ONE. After careful consideration, we feel that it has merit but does not fully meet PLOS ONE’s publication criteria as it currently stands. Therefore, we invite you to submit a revised version of the manuscript that addresses the points raised during the review process. Please submit your revised manuscript by Jul 05 2024 11:59PM. If you will need more time than this to complete your revisions, please reply to this message or contact the journal office at plosone@plos.org. Please include the following items when submitting your revised manuscript:A rebuttal letter that responds to each point raised by the academic editor and reviewer(s). You should upload this letter as a separate file labeled 'Response to Reviewers'.A marked-up copy of your manuscript that highlights changes made to the original version. You should upload this as a separate file labeled 'Revised Manuscript with Track Changes'.An unmarked version of your revised paper without tracked changes. You should upload this as a separate file labeled 'Manuscript'.

We look forward to receiving your revised manuscript.

Kind regards,

Sairah Hafeez Kamran, PhD

Academic Editor

PLOS ONE

2. To comply with PLOS ONE submissions requirements, in your Methods section, please provide additional information regarding the experiments involving animals and ensure you have included details on (1) methods of sacrifice, (2)  efforts to alleviate suffering.

“This work was supported by the Grants from National Natural Science Foundation of China [Nos. 81450016 and 81272786 to J.L.] and Natural Science Foundation of Liaoning Province, China [No. 2019-ZD-0650 to P.N. L.]”

4. PLOS requires an ORCID iD for the corresponding author in Editorial Manager on papers submitted after December 6th, 2016. Please ensure that you have an ORCID iD and that it is validated in Editorial Manager. To do this, go to ‘Update my Information’ (in the upper left-hand corner of the main menu), and click on the Fetch/Validate link next to the ORCID field. This will take you to the ORCID site and allow you to create a new iD or authenticate a pre-existing iD in Editorial Manager. Please see the following video for instructions on linking an ORCID iD to your Editorial Manager account: https://www.youtube.com/watch?v=_xcclfuvtxQ.

Reviewers' comments:

Reviewer's Responses to Questions

**Comments to the Author**

1. Is the manuscript technically sound, and do the data support the conclusions?

Reviewer #1: Yes

Reviewer #2: Partly

2. Has the statistical analysis been performed appropriately and rigorously? 

Reviewer #1: Yes

Reviewer #2: Yes

3. Have the authors made all data underlying the findings in their manuscript fully available?

Reviewer #1: Yes

Reviewer #2: Yes

4. Is the manuscript presented in an intelligible fashion and written in standard English?

Reviewer #1: Yes

Reviewer #2: No

5. Review Comments to the Author

Reviewer #1: Review Report:

The manuscript titled "Maggot extract accelerates skin wound healing of diabetic rats via enhancing STAT3 signaling" suggests an intriguing investigation into the potential therapeutic effects of maggot extract on wound healing, particularly in diabetic rats. The study likely delves into the mechanisms by which maggot extract may expedite the healing process, with a specific focus on its impact on STAT3 signaling pathways. The use of maggot extract for wound healing purposes represents a novel and potentially innovative approach, especially if the study provides evidence supporting its efficacy. Overall, the manuscript appears to present a promising study with implications for both basic research and clinical practice in the field of wound healing and diabetes management.

Note: I reviewed the file named (PONE-D-24-13961) downloaded from journal online portal and all my comments are according to the line numberings of the pdf file “PONE-D-24-13961”

Detailed Comments/Recommendations:

Title:

The title of the study, "Maggot extract accelerates skin wound healing of diabetic rats via enhancing STAT3 signaling," appears to be apt and engaging. It succinctly captures the main focus of the research while also hinting at the potential mechanisms involved. This descriptive and informative title is likely to attract readers' attention and convey the significance of the study's findings.

Abstract:

Background: Rewrite the starting line of the paragraph in a meaningful and in scientific representative way. The scientific name of Lucilia sericata must be italicized in the abstract and in everywhere of the manuscript.

Methods: The dose of the streptozotocin must be mentioned. Similarly, it is not mentioned that how the wounds were made in rats. Rats groupings is confusing. Please elaborate the groups according to their treatment protocols. Kindly briefly describe the information about wound formation.

Results: Rewrite the first line of this section and 14th, must be written in this way.

Conclusion: In objective the authors claimed to explore the molecular mechanism of healing but the conclusion of the study in the last of abstract is not exactly supporting the aims and objective.

Key Words: Authors used 4 out of 5 same words as mentioned in title. The key words should be different from the words used in title so that the reader may explore the study with different more focused words.

Note: The authors are requested to resolve the above mentioned points and rewrite the abstract in an appropriate comprehensive scientific language and give focused key words.

Introduction:

There is no coordination in starting both paragraphs of the introduction. It will be more suitable if authors write the introduction in sequence in term of following points; introduction of diabetes, its types, complications, wounds pathogenesis, treatment regimens available, and treatment failures.

Starting in 3rd paragraph of introduction, there is lacking about the introduction (family, genus, distribution, habitat etc) of Lucilia sericata.

The last paragraph of Introduction is well written.

Kindly add on the aims and objectives of the current study.

Materials and Methods:

Maggot Extract Preparation:

Lucilia cuprina should be written in italic as Lucilia cuprina. In the Material and Method, subsection Maggot Extract Preparation, Lucilia cuprina is mentioned while in abstract and introduction Lucilia sericata is discussed.

For extract preparation of Maggots, from where the larvae and its organism were identified? Kindly mention the details from where the organism was collected or obtained and give identification details as well.

Kindly describe the standard conditions at which the blowflies were maintained.

Kindly write full form of PBS.

Kindly remove the space between µ m. it should be written as µm.

rhEGF details are missing. From where it was obtained? Give specification and supplier details.

What are the dressing reagents? Give elaboration at the end of subsection Skin Wound Model and Treatment.

In subsection Wound Examination, the critera for wound healing and its markers etc are not mentioned.

Wound edge biopsy: the procedure of taking biopsy is not mentioned.

Results:

Establishment of diabetic rat model: “The typical symptoms of diabetes include the increased drinking, eating and urinating, which appeared to the rats on the third day after streptozotocin injection. As shown in Fig 2, the model rats but not the normal ones showed lost weight, and the fasting blood glucose levels of normal SD rats were less than 7.0 mmol/L, and in streptozotocin-treated rats were distinctly increased in the range from 25.6 to 27.3 mmol/L and 26.5 ± 0.8 mmol/L in average” This paragraph should be rewritten in correct meaningful way.

Active regeneration of maggot extract- and rhEGF-treated wounds: Rewrite this sentence: “HE and Masson’s trichrome stainings were performed on the whole layer skin wound tissues biopsied from each experimental group on the 14th day to analyze re-epithelialization and granulation tissue formation”.

Discussion:

Over all the discussion is well written and quite satisfactory.

On the 14th day, should be written as; On the 14th day.

It is written in discussion that; “The beneficial effects of maggots on wound healing have long been recognized[27,28] and its extract contains alkaloids, oils, polypeptides and proteins that promote local angiogenesis and cell proliferation [9]”. Please mention the name of any oil, alkaloid, polypeptide etc which has been isolated so from maggot and is used for wound healing.

The authors are requested to check out all the in text citation that there must be space between text and citation. At most of places it is written as; recognized[27,28], It should be like that; recognized [27,28].

There is lacking any scientific evidence about the key constituent of maggot responsible for the wound healing.

Conclusion:

It should be rewritten by keeping in view the aims and objective of study and proceeding towards their achievement.

References:

Reference no 31: Mol Cell Endocrinol. Why the authors write journal name in italic. Kindly remove the italics.

The authors are advised to review all the references for the strict adherence of the uniformity of Journal’s references style.

Concluding Remarks:

The paper necessitates the aforementioned corrections, inclusive of rectifying any errors highlighted, coupled with a meticulous proofreading by a proficient language expert to refine its English language usage. Upon addressing these amendments, the paper will meet the required standards for acceptance.

Reviewer #2: The study is good contribution to the diabetic wound healing research and will help others once published.

I would suggest:

Enhance the overall quality of manuscript.

Better phrases can be used.

Abbreviations need to be explained.

Ensure soundness of data, results and their presentation.

A lot is needed to be done before making it to final publishing.

6. PLOS authors have the option to publish the peer review history of their article (what does this mean?). If published, this will include your full peer review and any attached files.

Reviewer #1: **Yes: **Dr. Aamir Mushtaq

Reviewer #2: No

---

## [Author Response · Author response to Decision Letter 0]

29 Jun 2024

Rebuttal Letter

1. To comply with PLOS ONE submissions requirements, in your Methods section, please provide additional information regarding the experiments involving animals and ensure you have included details on (1) methods of sacrifice, (2) efforts to alleviate suffering.

Response: Thanks. To alleviate suffering, the rats were first anesthetized with ketamine and xylazine intraperitonealy and then sacrificed through cervical dislocation performed by appropriately trained and competent personnel at the end of the experiment, which was described in Methods section and highlighted in red as “The rats were anesthetized and sacrificed through cervical dislocation performed by appropriately trained and competent personnel at the end of the experiment. The skin tissues in the size of 0.3 × 0.3 × 0.2 cm .……” 

2. Please change the highlighted text in your manuscript with black text.

Response: Thanks. The highlighted text has been changed to black in the manuscript. 

3. "We note that your Data Availability Statement is currently as follows: [All relevant data are within the manuscript and its Supporting Information files.]

If there are ethical or legal restrictions on sharing a de-identified data set, please explain them in detail (e.g., data contain potentially sensitive information, data are owned by a third-party organization, etc.) and who has imposed them (e.g., an ethics committee). Please also provide contact information for a data access committee, ethics committee, or other institutional body to which data requests may be sent. If data are owned by a third party, please indicate how others may request data access."

Response: According to the requirements above, we have uploaded the raw data named as S2 Raw data, S3 Raw data and S4 Raw data in Supporting Information files.

Reviewer #1: Review Report:

The manuscript titled "Maggot extract accelerates skin wound healing of diabetic rats via enhancing STAT3 signaling" suggests an intriguing investigation into the potential therapeutic effects of maggot extract on wound healing, particularly in diabetic rats. The study likely delves into the mechanisms by which maggot extract may expedite the healing process, with a specific focus on its impact on STAT3 signaling pathways. The use of maggot extract for wound healing purposes represents a novel and potentially innovative approach, especially if the study provides evidence supporting its efficacy. Overall, the manuscript appears to present a promising study with implications for both basic research and clinical practice in the field of wound healing and diabetes management.

Note: I reviewed the file named (PONE-D-24-13961) downloaded from journal online portal and all my comments are according to the line numberings of the pdf file “PONE-D-24-13961”

Detailed Comments/Recommendations:

Title: The title of the study, "Maggot extract accelerates skin wound healing of diabetic rats via enhancing STAT3 signaling," appears to be apt and engaging. It succinctly captures the main focus of the research while also hinting at the potential mechanisms involved. This descriptive and informative title is likely to attract readers' attention and convey the significance of the study's findings.

Abstract: Background: Rewrite the starting line of the paragraph in a meaningful and in scientific representative way. The scientific name of Lucilia sericata must be italicized in the abstract and in everywhere of the manuscript.

Response: Thank you for your suggestion. We have rewritten the starting line of the “Background” paragraph and highlighted in red in ABSTRACT section as: Diabetic skin wound is a complex problem due to the disruption of normal repairing program and lack of effective remedy. The scientific name of Lucilia sericata have been corrected as italicized in the abstract and in everywhere of the manuscript.

Methods: The dose of the streptozotocin must be mentioned. Similarly, it is not mentioned that how the wounds were made in rats. Rats groupings is confusing. Please elaborate the groups according to their treatment protocols. Kindly briefly describe the information about wound formation.

Response: 

1) According to your suggestion, the dose of the streptozotocin has been added in the “Methods” paragraph and highlighted in red as: The rat fasting blood glucose values were ≧16.7 mmol/L 72 hours after intraperitoneal streptozotocin (60 mg/kg body weight) injection. 

2) The groups in the “Methods” paragraph and highlighted in red as: The rats were divided into five groups (n = 10/group): normal group (normal SD rats without any treatment), diabetic blank group (the diabetic rats without any treatment), Vaseline group (the diabetic rats dressed with Vaseline), rhEGF (recombinant human epidermal-growth-factor) group (the diabetic rats dressed with a mixture of Vaseline and 200 μg/g rhEGF), M.E. group (the diabetic rats dressed with a mixture of Vaseline and 150 μg/ml maggot extract) .

3) The information about wound formation has been described and highlighted in red in the “Methods” paragraph as: The round open wounds (1.0 cm in diameter) down to the muscle fascia were made on both sides of rat dorsa by removing the skin layer (epidermis and dermis) and were daily photographed for calculating their healing rates.

Results: Rewrite the first line of this section and 14th, must be written in this way.

Response: The first line of this section was rewritten as “The average wound healing rates on the 14th day were 91.7 % in the normal, 79.6% in M.E., 71% in rhEGF, 55.1% in vaseline and 43.3% in the diabetes blank group”, which was highlighted in red. 

Conclusion: In objective the authors claimed to explore the molecular mechanism of healing but the conclusion of the study in the last of abstract is not exactly supporting the aims and objective.

Response: Thank you for this valuable suggestion. The “Conclusion” has been rewritten as “Maggot extract would be an alternative and/or adjuvant candidate for the better management of diabetic skin wounds because of its activity in enhancing STAT3 activation.”

Key Words: Authors used 4 out of 5 same words as mentioned in title. The key words should be different from the words used in title so that the reader may explore the study with different more focused words.

Response: According to your advice, Key Words have been changed to “Lucilia sericata larvae extract, diabetic skin wound, streptozotocin, signal transducer and activator of transcription 3, wound healing related genes”.

Note: The authors are requested to resolve the above mentioned points and rewrite the abstract in an appropriate comprehensive scientific language and give focused key words.

Introduction:

There is no coordination in starting both paragraphs of the introduction. It will be more suitable if authors write the introduction in sequence in term of following points; introduction of diabetes, its types, complications, wounds pathogenesis, treatment regimens available, and treatment failures.

Response: According to your comment. we have integrated starting both paragraphs of the introduction into one paragraph in the order of introduction of diabetes, its types, complications, wounds pathogenesis, treatment regimens available, and treatment failures, which is shown in Introduction part of text in red: “Diabetes is a chronic metabolic disease, which has become a growing international health concern [1, 2]. Diabetes (type 1 and 2) involves many complications, and the most common one is the delayed healing and even nonunion of acute and chronic skin wounds [3]. Multiple factors or molecular machinaries responsible for wound healing are altered in diabetic individuals, leading to the reduced regenerative activity of skin wounds. It has been proposed that the local peripheral nerve and vascular alterations are the main reasons for the difficulty of skin wound healing [4]. Current options for wound management of diabetic patients are: 1) strict control of blood glucose; 2) application of vasodilators to improve microvascular circulation; 3) antibacterial and anti-inflammatory treatments; 4) surgical debridement and 5) local nursing and health care [5]. Although the above measures include both the whole and local treatments, the incidence of diabetic wounds and the disability rate caused by them remain almost unchanged, indicating the unsatisfactory efficacy of the above approaches [6]. Therefore, it would be necessary to alleviate the influence of pathogenic factors and meanwhile to accelerate the wound healing. Because of the complexity of the pathogenic factors of diabetic wounds, the new agent (drug) containing beneficial components for neovascularization and cell proliferation would be helpful.”

Starting in 3rd paragraph of introduction, there is lacking about the introduction (family, genus, distribution, habitat etc) of Lucilia sericata.

Response: According to your suggestion, the introduction of Lucilia sericata have been added in the “Introduction” part and highlighted in red as: “Lucilia sericata (Meigen 1826, Diptera: Calliphoridae) is a widespread blow fly species.........”.

The last paragraph of Introduction is well written.

Kindly add on the aims and objectives of the current study.

Response: Thank you for your comments. the aims and objectives of the current study have been added at the end of the “Introduction” part and highlighted in red as: “The current study aims to address the therapeutic effects of maggot extract on diabetic skin wound and its molecular mechanism.”

Materials and Methods:

Maggot Extract Preparation:

Lucilia cuprina should be written in italic as Lucilia cuprina. In the Material and Method, subsection Maggot Extract Preparation, Lucilia cuprina is mentioned while in abstract and introduction Lucilia sericata is discussed.

Response: Thank you for your comments. Lucilia cuprina have been corrected to Lucilia sericata.

For extract preparation of Maggots, from where the larvae and its organism were identified? Kindly mention the details from where the organism was collected or obtained and give identification details as well.

Kindly describe the standard conditions at which the blowflies were maintained.

Response: Thank you for your comments. The related information have been added in the text as “Lucilia sericata blowflies were obtained from BIOWIM Technology Development Co., Ltd (Dalian, China) and maintained in a closed container with a net at a constant temperature of about 22℃. Their larvae were collected when they were reared to later-second or early-third stages. The collected maggots were rinsed three times with autoclaved pure water, placed in 3.5% formaldehyde normal saline solution for 5 minutes, in 2% H2O2 for 3 minutes, in 1% hydrochloride acid for 5 minutes and finally subjected to three washes with autoclaved pure water. ”



Kindly write full form of PBS.

Response: Thank you for your comments. The full form of PBS is phosphate buffered saline, which has been added in the text as “ ……putting the frozen sections into the test tubes containing pH7.5 phosphate buffered saline (PBS). ”

 Kindly remove the space between µ m. it should be written as µm.

Response: Thank you for your careful reading. The typo has been corrected.

rhEGF details are missing. From where it was obtained? Give specification and supplier details.

What are the dressing reagents? Give elaboration at the end of subsection Skin Wound Model and Treatment.

Response: Thanks for your comments. rhEGF was purchased from Guilin Pavay Gene Pharmaceutical Co., Ltd (Guilin, China). rhEGF details and the dressing reagents have been added to the text as “ ……Group 4, diabetic rats dressed with a mixture of Vaseline and 200 μg/g recombinant epidermal growth factor (Pavay Gene Pharmaceutical, Guilin, China, rhEGF group); Group 5, the diabetic rats dressed with a mixture of Vaseline and 150 μg/ml maggot extract (M.E. group). The experimental treatments lasted for 14 days by daily dressing the reagents mentioned above. ”

In subsection Wound Examination, the critera for wound healing and its markers etc are not mentioned.

Response: Thanks for your suggestion. The way to evaluate wound healing has been described in the text as “The wound areas were measured and the wound healing rates were calculated to evaluate the wound healing in differen experimental groups.” 

Wound edge biopsy: the procedure of taking biopsy is not mentioned.

Response: Thanks for your suggestions. The procedure of taking biopsy has been described in “Wound edge biopsy” subsection as “The rats were anesthetized and sacrificed through cervical dislocation performed by appropriately trained and competent personnel at the end of the experiment. The skin tissues in the size of 0.3 × 0.3 × 0.2 cm were biopsied from the wound margins of the five experimental groups. Half of the tissues was snap-frozen in liquid nitrogen……”

Results:

Establishment of diabetic rat model: “The typical symptoms of diabetes include the increased drinking, eating and urinating, which appeared to the rats on the third day after streptozotocin injection. As shown in Fig 2, the model rats but not the normal ones showed lost weight, and the fasting blood glucose levels of normal SD rats were less than 7.0 mmol/L, and in streptozotocin-treated rats were distinctly increased in the range from 25.6 to 27.3 mmol/L and 26.5 ± 0.8 mmol/L in average” This paragraph should be rewritten in correct meaningful way.

Response: According to your suggestion, this paragraph has been rewritten in the text as “The typical symptoms of diabetes include polydipsia, polyphagia and polyuria, which was observed in rats on the third day after streptozotocin injection. As shown in Fig 2, the diabetic rats but not the normal ones showed weight loss, whereas the fasting blood glucose levels of normal rats were less than 7.0 mmol/L, and in streptozotocin-treated rats were distinctly increased (ranging from 25.6 to 27.3 mmol/L). The difference of fasting blood glucose levels between the normal and streptozotocin-treated rats was statistically significant (P < 0.01; Table 1).”

Active regeneration of maggot extract- and rhEGF-treated wounds: Rewrite this sentence: “HE and Masson’s trichrome stainings were performed on the whole layer skin wound tissues biopsied from each experimental group on the 14th day to analyze re-epithelialization and granulation tissue formation”.

Response: Thanks for your suggestion. This sentence has been rewritten in the text as “To investigate wound regeneration, re-epithelialization and granulation tissue formation were checked by HE and Masson’s trichrome staining performed on the whole layer skin wound tissues biopsied from each experimental group on the 14th day. ”

Discussion:

Over all the discussion is well written and quite satisfactory.

On the 14th day, should be written as; On the 14th day.

Response: Thanks for your comments. The sentence has been corrected in the text as “The wounds of normal rats were almost healed (healing rates: 91.7% ± 3.7%), while the wounds of the corresponding diabetic group healed partly (healing rates: 43.3% ± 2.3

---

## [Decision Letter · Decision Letter 1]

31 Jul 2024

PONE-D-24-13961R1Maggot extract accelerates skin wound healing of diabetic rats via enhancing STAT3 signalingPLOS ONE

Dear Dr. Liu,

Thank you for submitting your manuscript to PLOS ONE. After careful consideration, we feel that it has merit but does not fully meet PLOS ONE’s publication criteria as it currently stands. Therefore, we invite you to submit a revised version of the manuscript that addresses the points raised during the review process.

We look forward to receiving your revised manuscript.

Kind regards,

Sairah Hafeez Kamran, PhD

Academic Editor

PLOS ONE

Journal Requirements:

Reviewers' comments:

Reviewer's Responses to Questions

**Comments to the Author**

1. If the authors have adequately addressed your comments raised in a previous round of review and you feel that this manuscript is now acceptable for publication, you may indicate that here to bypass the “Comments to the Author” section, enter your conflict of interest statement in the “Confidential to Editor” section, and submit your "Accept" recommendation.

Reviewer #1: All comments have been addressed

Reviewer #3: (No Response)

2. Is the manuscript technically sound, and do the data support the conclusions?

Reviewer #1: Yes

Reviewer #3: (No Response)

3. Has the statistical analysis been performed appropriately and rigorously? 

Reviewer #1: Yes

Reviewer #3: (No Response)

4. Have the authors made all data underlying the findings in their manuscript fully available?

Reviewer #1: Yes

Reviewer #3: (No Response)

5. Is the manuscript presented in an intelligible fashion and written in standard English?

Reviewer #1: Yes

Reviewer #3: (No Response)

6. Review Comments to the Author

Reviewer #1: The revised version of paper entitled "The manuscript titled "Maggot extract accelerates skin wound healing of diabetic rats via enhancing STAT3 signaling" has been reviewed by me and it has been found that the authors have resolved all the mentioned technical issues. However, at certain points there are few English language related errors. For the well balanced presentation of the paper i would like to advise the authors to get this paper proof read by a proficient language expert to refine its English language usage. From me side it is accepted for the publication as the authors have successfully addressed all of my comments.

Reviewer #3: I greatly appreciate considering me as a reviewer for the manuscript PONE-D-24-13961R1.

1) In the section of Skin wound model and treatments, author should add reference that illustrate using vaseline as dressing for diabetic wound.

2) All figures resolution should be improved.

3) Quantity of Maggot extract should be mentioned

7. PLOS authors have the option to publish the peer review history of their article (what does this mean?). If published, this will include your full peer review and any attached files.

Reviewer #1: **Yes: **Dr. Aamir Mushtaq

Reviewer #3: No

---

## [Author Response · Author response to Decision Letter 1]

13 Aug 2024

Rebuttal Letter

Journal Requirements:

Response: According to the requirements above, we have checked carefully all the references one by one to ensure that each is complete and correct and have not cited any papers that have been retracted.

Reviewer #1: The revised version of paper entitled "The manuscript titled "Maggot extract accelerates skin wound healing of diabetic rats via enhancing STAT3 signaling" has been reviewed by me and it has been found that the authors have resolved all the mentioned technical issues. However, at certain points there are few English language related errors. For the well balanced presentation of the paper i would like to advise the authors to get this paper proof read by a proficient language expert to refine its English language usage. From me side it is accepted for the publication as the authors have successfully addressed all of my comments.

Response: Thanks for your comments. We have got this paper read carefully and corrected the grammatical errors and typos found in the secondly revised manuscript.

Reviewer #3: I greatly appreciate considering me as a reviewer for the manuscript PONE-D-24-13961R1.

1) In the section of Skin wound model and treatments, author should add reference that illustrate using vaseline as dressing for diabetic wound.

Response: Thanks for your suggestion. We have added the reference that illustrate using vaseline as dressing for diabetic wound in the section of Skin wound model and treatments and highlighted it in red as “……Group 3, diabetic rats dressed with Vaseline (Rhawn, Shanghai, China, Vaseline group) [11]; Group 4, diabetic rats dressed with a mixture of Vaseline [11] and 200 μg/g recombinant epidermal growth factor (Pavay Gene Pharmaceutical, Guilin, China, rhEGF group); Group 5, the diabetic rats dressed with a mixture of Vaseline [11] and 150 μg/ml maggot extract (M.E. group)” .

2) All figures resolution should be improved.

Response: Yes. We have improved the resolution of all figures.

3) Quantity of Maggot extract should be mentioned.

Response: Thank you for your suggestion. The quantity of maggot extract and the quantitative method have been added in the section of Skin wound model and treatments and highlighted in red as “……Group 5, the diabetic rats dressed with a mixture of Vaseline [11] and 150 μg/ml maggot extract (M.E. group). The experimental treatments lasted for 14 days by daily dressing the reagents mentioned above. Maggot extract was quantified by Ultraviolet spectrophotometer-based calculation of protein concentration. ”

---

## [Decision Letter · Decision Letter 2]

21 Aug 2024

Maggot extract accelerates skin wound healing of diabetic rats via enhancing STAT3 signaling

PONE-D-24-13961R2

Dear Dr. Liu,

We’re pleased to inform you that your manuscript has been judged scientifically suitable for publication and will be formally accepted for publication once it meets all outstanding technical requirements.

Kind regards,

Sairah Hafeez Kamran, PhD

Academic Editor

PLOS ONE

Reviewers' comments:

Reviewer's Responses to Questions

**Comments to the Author**

1. If the authors have adequately addressed your comments raised in a previous round of review and you feel that this manuscript is now acceptable for publication, you may indicate that here to bypass the “Comments to the Author” section, enter your conflict of interest statement in the “Confidential to Editor” section, and submit your "Accept" recommendation.

Reviewer #3: (No Response)

2. Is the manuscript technically sound, and do the data support the conclusions?

Reviewer #3: (No Response)

3. Has the statistical analysis been performed appropriately and rigorously? 

Reviewer #3: (No Response)

4. Have the authors made all data underlying the findings in their manuscript fully available?

Reviewer #3: (No Response)

5. Is the manuscript presented in an intelligible fashion and written in standard English?

Reviewer #3: (No Response)

6. Review Comments to the Author

Reviewer #3: (No Response)

7. PLOS authors have the option to publish the peer review history of their article (what does this mean?). If published, this will include your full peer review and any attached files.

Reviewer #3: No

---

## [Editor Report · Acceptance letter]

27 Aug 2024

PONE-D-24-13961R2 

PLOS ONE

Dear Dr. Liu, 

I'm pleased to inform you that your manuscript has been deemed suitable for publication in PLOS ONE. Congratulations! Your manuscript is now being handed over to our production team.

Kind regards, 

on behalf of

Dr. Sairah Hafeez Kamran 

Academic Editor

PLOS ONE